# Real Time Ultrasound-Guided Thoracic Epidural Catheterization with Patients in the Lateral Decubitus Position without Flexion of Knees and Neck: A Preliminary Investigation

**DOI:** 10.3390/jcm11216459

**Published:** 2022-10-31

**Authors:** Yuexin Huang, Tingting Li, Tianhong Wang, Yanhuan Wei, Liulin Xiong, Tinghua Wang, Fei Liu

**Affiliations:** 1Department of Anesthesiology, Institute of Neurological Disease, West China Hospital, Sichuan University, Chengdu 610044, China; 2Graduate School of Education, Beijing Foreign Studies University, Beijing 100039, China; 3Department of Anesthesiology, Affiliated Hospital of Zunyi Medical University, Zunyi 563000, China

**Keywords:** thoracic epidural puncture and catheterization, position, ultrasound image

## Abstract

**Objectives:** For some patients, such as pregnant women, it can be difficult to maintain the ideal “forehead to knees” position for several minutes for epidural catheter placement. We conducted this study to investigate the feasibility of real-time ultrasound-guided (US) epidural catheterization under a comfortable lateral position without flexion of knees and neck. **Materials and Methods:** 60 patients aged 18-80 years with a body mass index of 18-30 kg/m^2^ after general surgery were included. In a comfortable left lateral position, thoracic epidural catheterization was performed under real-time US for postoperative analgesia. The visibility of the neuraxial structures, procedural time from needle insertion to loss of resistance in the epidural space, the number of needle redirections, success rate of epidural catheter placement and postoperative analgesic effect were recorded. **Results:** In the paramedian oblique sagittal view, the well visible of vertebral lamina, intervertebral space and posterior complex under ultrasound were as high as 93.33%, 81.67% and 70.00%, respectively. The success rate of thoracic epidural catheterization was as high as 91.67%, and the satisfactory postoperative analgesic effect was 98.2% for patients without nausea, pruritus and other discomfort. **Discussion:** Thoracic epidural catheterization with patients in the lateral position without flexion of knees and neck under real time ultrasound guidance has a high success rate and strong feasibility. This visual manipulation makes epidural catheterization not only “easier” to perform, but also reduces the requirements of the procedure.

## 1. Key Points Summary

Question: To improve patients’ comfort, can we perform epidural space puncture and catheterization without maintaining the “ideal position”?

Findings: Thoracic epidural space catheterization (T6-L1) to patients without flexion of knees and neck (comfortable left lateral position) under ultrasound guidance has a high success rate (91.67%, 55/60) and satisfactory postoperative analgesic effect (98.18%, 54/55), and no patients experienced nausea, pruritus and other discomfort.

Meaning: Under a comfortable lateral position, ultrasound-guided epidural space puncture and catheterization is feasible.

## 2. Introduction

Epidural space puncture and catheterization can be used for intraoperative anesthesia and postoperative analgesia [1]. The lateral position is one of the most commonly used positions for spinal and epidural anesthesia, in which the patients have to flex their thighs on their abdomen and flex their neck to allow the forehead to be as close possible to the knees (“forehead to knees”, Figure 1 Left) [2]. This position allows the operator to obtain the presence of palpable spinal apophyses and easy determine the needle puncture space, and this influences the success rate of epidural catheterization [3,4]. However, it is difficult for some patients to maintain this position for several minutes without movement, which has been identified as a risk factor for dural puncture during epidural anesthesia [5]. Ultrasound (US) imaging can now be a valuable tool to view tissue and structures like nerve and spine anatomy, which could improve the clinical efficacy of epidural catheter placement, reduce the risk of failed or traumatic procedures, and decrease the number of attempts and redirections of the needle [6,7,8,9,10,11]. Until now, there have been no reports on whether the US has an advantage in epidural catheter placement in patients who cannot maintain the “forehead to knees” ideal position. Thus, we planned to do thoracic epidural catheter placement under real time US-guidance for the patients in the lateral decubitus position without flexion of knees and neck in order to explore the possibility of thoracic epidural catheterization under ultrasound in a comfortable position, and to provide the relevant imaging data of thoracic epidural catheterization under ultrasound.

## 3. Materials and Methods

### 3.1. Ethic

A single-center case series study was performed. Ethical approval for the study was provided by the Ethics Committee of our institution (West China Biomedical Ethics Committee, Sichuan University. No. 2016 (332). 18 February 2019.), and written patient informed consent was obtained from all patients. This study was retrospectively registered on ClinicalTrials.gov (ChiCTR2100054727). All procedures in this study were in accordance with the ethical standards of the Helsinki Declaration and the international ethical guidelines for human biomedical research. This manuscript adheres to the applicable STROBE guidelines.

### 3.2. Patients

A total 60 patients aged 18–80 with American Society of Anesthesiologists (ASA) classification of I or II, who were scheduled to undergo abdominal surgeries under GA, were enrolled in the study from 1 September to 30 November 2020 at the West China Hospital of Sichuan University. Patients with neurological disorders, seizures, history of spine surgeries or deformities, a history of local anesthetic allergy, local site infection, or coagulopathies were excluded. Epidural catheterization was performed by Fei Liu, the corresponding author of this article, who is skilled in ultrasound guiding epidural puncture. The level of epidural puncture was determined by the surgical incision. Ultrasound imaging was completed using an M-turbo curved probe of 2–5 MHz (FUJIFILM Sonosite, Inc. Bothell, WA, USA). All of the patients were followed for up to 1 week after surgery. If patients had adverse reactions, patients were observed until recovery.

### 3.3. Operational Process

Epidural puncture and catheterization were performed preoperatively. Intravenous access and standard ASA monitors (pulse oximetry, electrocardiogram, and noninvasive blood pressure) were established prior to epidural puncture. All of the patients lied in a comfortable position of left lateral decubitus (they were not required to flex their thighs on their abdomen and flex their neck), as shown in Figure 1 right). The paramedian oblique sagittal plane of the target epidural space was identified and marked by applying the probe from the 12th rib in the pre-procedure scanning and using the counting up method [12]. (The scanning procedure could be found in the Appendix A). After aseptic skin disinfection, the probe was firmly held in the marked position by the operator’s left hand (Figure 2 Left) and applied at the congruent thoracic level to obtain a paramedian longitudinal view of the spine. A 18G Tuohy needle from the caudal of the probe was advanced to the interlaminar space under real-time ultrasound guidance using an in-plane approach until the needle tip reached the posterior part of the ligamentum flavum-dura mater complex by the operator’s right hand (Figure 2 Right). The ultrasound probe was then left aside and the two hands of the single operator held the needle to continue the procedure until the epidural space was identified with loss of resistance to air [4] Finally, the epidural catheter was advanced 4–5 cm inside the epidural space and secured at the back with transparent dressings. (The puncturing procedure can be found in the operating Appendix A).

### 3.4. Data Collection

(1) Characteristics of patients, including gender, age, height, weight, and body mass index (BMI), history of hypertension or chronic bronchitis. (2) The operation information. (3) The ultrasound visibility of neuraxial structures at the punctured interlaminar space [13] (including vertebral lamina, intervertebral space, dural sac, anterior complex and posterior complex) were also assessed by an independent observer using a 4-point Likert scale (0-point = not visible, 1-point = hardly visible, 2-point = well visible, 3-point = very well visible), and the total ultrasound visibility score (UVS, maximum score possible = 15) for each patient was determined. (4) Vertical and oblique distances from the skin to the posterior dura (Figure 3A,B) and width of the interlaminar space were measured (Figure 3C). (5) Puncture Outcome, including time for the US pre-procedure scanning and epidural space puncture (from the start of the skin puncture to the confirmation of the needle tip in the epidural space), success rate of epidural puncture and catheterization, analgesia effect, and adverse reaction were recorded. Mild adverse effects included dural puncture, bloody tap, paresthesia during needle insertion or catheterization and severe adverse effects contained total spinal anesthesia, epidural hematoma or abscess related to epidural puncture.

### 3.5. Statistical Analysis

The data was analyzed using SPSS (Version 21.0, IBM, Chicago, IL, USA). Measurement data were described as mean ± standard deviation (Std), or median (interquaternary interval, IQR). The enumeration data were described by frequency and composition ratio. The correlation between patient characteristics, ultrasonic imaging and epidural puncture time was analyzed. The influence degree of each influencing factor on the epidural puncture time was then analyzed. According to our institution’s previous experience with ultrasound-guided catheterization, the success rate is approximately 90%. Thus, when we calculated the sample size, we assumed the success rate to be 90%, the confidence interval width to be 16%, the allowable error range to be 8%, the dropout rate to be 10%, and α = 0.05. By using PASS 2021 (Version 21.0.3), a sample size of 54 patients were calculated, and we choose 60 patients as the final sample size.

## 4. Results

Real-time US-guided epidural space catheters were successfully finished in 55 patients of the 60 patients (41 male and 19 female) under a comfortable left lateral decubitus, with a mean age of 57.44 years and BMI of 21.88 kg/m^2^. Time of pre-procedure scanning and puncture was 44.78 ± 24.81 s and 119.37 ± 116.70 s. During the procedure, one epidural puncture was cancelled due to poor visibility of the target interlaminar space during preprocedure scanning; the catheterization in four patients was cancelled because of one catheter expected into the blood vessel, while the other three patients reported transient paresthesia. Of the 55 patients with successful puncture and catheterization, 98.18% reported a satisfactory analgesic effect. No complication related to epidural puncture was found until discharge.

### 4.1. Basic Characteristics of Patients

Among the 60 patients, 68.33% (41/60) were male and 31.67% (19/60) were female. Patients ranged in age from 29 to 80 years (57.44 ± 11.36 years), in which 29 patients were older than 60 years old (29/59, 49.15%). The height of the patients was 163.91 ± 6.82 cm (cm), and the weight was 59.30 ± 9.26 kilograms (kg). The BMI was between 18.00 and 30.00 kg/m^2^ (21.88 ± 3.07 kg/m^2^). Among them, patients with overweight body (BMI ≥ 24 kg/m^2^) accounted for 21.82% (12/55) (Table 1).

### 4.2. Complicated Diseases of Patients

Among the 60 included patients, there were four patients who had complications of hypertension (6.67%), four patients who experienced chronic bronchitis (6.67%), and two patients who showed abnormal ECGs in their preoperative examination (3.33%) (Table 1). In the two patients with abnormal ECG, one patient had sinus bradycardia, and the other patient reported premature ventricular beats.

### 4.3. Information on Surgical Treatment of Patients

Among the 60 cases, four cases underwent chest surgery (6.67%, all of which were via lobotomy) and 56 cases underwent abdominal or pelvic surgery (93.33%) (stomach or jejunum: 31/56, 55.36%; colorectum: 15/56, 26.79%; others: 10/56, 17.86%). Among the 60 cases, 50 cases underwent laparoscopic or thoracoscopic surgery (83.33%), and 10 cases underwent open surgery (16.67%).

### 4.4. Ultrasonic Imaging

During this observation period, ultrasound localization required 44.66 ± 25.72 s per case, and all was completed within 2 min (Figure 4A). The vertical distance from skin to dura was 3.75 ± 0.70 cm (a range of 2.09–6.10 cm). The oblique depth was 5.62 ± 0.59 cm (a range of 3.68–6.85 cm). Intervertebral space distance ranged from 0.31 to 1.55 cm (Mean ± Std: 0.70 ± 0.23 cm) (Figure 4B).

According to the visibility of the neuraxial structures (Table 2), the UVS of the included cases was measured to be 10.73 ± 5.58 (95% CI: 9.29–12.18, median 10.00, IQR 9.75, min 1.00, max 25.00) (Figure 4C). Among these structures, the proportion of vertebral lamina imaging visible was the highest (93.33%, well visible: 21/60, 35%; very well visible: 35/60, 58.33%), followed by intervertebral space (81.67%, well visible: 21/60, 35%; very well visible: 28/60, 46.67%) and posterior complex (70.00%, well visible: 27/60, 45%; very well visible: 15/60, 25.00%). The dural sac (45.00%, well visible: 16/60, 26.67%; very well visible: 11/60, 18.33%) and the anterior complex (40.00%, well visible: 13/60, 21.67%; very well visible: 11/60, 18.33%) showed poorly under ultrasound (Figure 4D).

### 4.5. Results of Epidural Catheterization

Of the 60 cases observed, only one patient abandoned the epidural puncture due to poor ultrasound imaging results, while the remaining patients successfully completed the epidural needle puncture (59/60, 98.33%). During the procedure, there was one patient that experienced a vascular puncture and three patients who experienced transient paresthesia; therefore, 55 of 60 patients were successfully catheterized (55/60, 91.67%). Under ultrasound guidance, one patient underwent epidural puncture at thoracic vertebra (T) 6-7 (1.67%), two patients at T 7-8 (3.39%), 35 patients at T 8-9 (59.32%), five patients at T 9-10 (8.47%), one patient at T 10-11 (1.69%), four patients at T 11-12 (6.78%), and 11 patients at T 12-lumbar vertebra (L) 1 (18.64%) (Figure 5A). The time to complete an epidural puncture was 71 (70) s (95% CI: 87.82–150.92, mean 119.37, Std 116.70, min 10.56, max 636.00) (Figure 5B). The proportion of epidural puncture time under 90 s was 53.33% (32/55). Among the patients with successful puncture, 47 patients were successfully punctured at the first puncture (47/59, 81.03%), seven patients were punctured by the second attempt (7/59, 12.07%), and five patients were successfully punctured by the third attempt (5/59, 8.47%) (Figure 5C). The puncture depth of the needle ranged from 3.00 to 8.00 (Figure 5D).

Among the 59 patients with successful puncture, one patient gave up catheterization because of puncture into the blood vessel (1.69%), and three patients gave up catheterization because of neuroparesthesia (3/59, 5.08%) (Figure 5E). Among the 55 patients with successful catheterization, 96.36% (53/55) had the catheterization completed successfully in only one attempt, and 3.64% (2/55) needed two attempts (Figure 5F). The distance from catheter tip to skin was 11.41 ± 1.21 cm (95% CI: 11.07–11.75, median 11.50, IQR 1.00, min 7.50, max 13.00), and the depth of the epidural space was between 2 and 8 cm (Mean ± Std: 4.72 ± 0.81cm, 95% CI: 4.49–4.94, Median 5, IQR 0.50) (Figure 5G).

### 4.6. Analgesic Effect and Complication

Of the 60 patients who underwent ultrasound-guided epidural catheterization, 55 (91.67%) completed the procedure successfully. Only one patient showed poor unilateral analgesic effect (1/55, 1.82%), and the remaining 54 patients reported satisfactory analgesic effect (54/55, 98.18%). No complication (dural puncture, bloody tap, total spinal anesthesia, epidural hematoma or abscess) was found before discharge.

### 4.7. Correlation Analysis Results

The time of ultrasound-guided epidural catheterization has no correlation with the patients’ sex (*p* = 0.150), age (*p* = 0.460), height (*p* = 0.096), weight (*p* = 0.834), BMI (*p* = 0.354), puncture plane (*p* = 0.816), ultrasound localization (*p* = 0.956), vertical distance from skin to dura (*p* = 0.195), UVS (*p* = 0.161), or visibility of several neuraxial structures: vertebral lamina (*p* = 0.282), intervertebral space (*p* = 0.160), and posterior complex (*p* = 0.163).

However, there are correlations among distance to the intervertebral space (*p* = 0.010, r = −0.358), oblique distance from skin to dura (*p* = 0.012, r = 0.338), and visibility of other neuraxial structures: dural sac (*p* = 0.006, r = −0.367), and anterior complex (*p* = 0.003, r = −0.395).

## 5. Discussion

This study is the first study to report that real-time US guidance could facilitate middle and low thoracic epidural puncture for the patients in the left lateral decubitus position without using the forehead to knees position. In this observational study of the 60 patients, 59 patients had a successful epidural puncture. Among them, 55 patients (91.67%) successfully completed epidural catheterization without any related adverse reactions. Except for one patient who had only good unilateral analgesia, the remaining 54 patients had satisfactory bilateral analgesia after surgery. During the process of catheterization, five patients withdrew from the operation due to contact with blood vessels or paresthesia. The visibility of the vertebral lamina, intervertebral space and posterior complex in the paramedian oblique sagittal view were all good at T6-L1 level in the patients without forehead to knees under ultrasound. And epidural catheterization to patients without flexion of knees and neck under ultrasound guidance has a high success rate and strong feasibility. In addition, this study provided detailed ultrasound data for the thoracic epidural puncture.

### 5.1. Real-Time Ultrasound-Assisted Thoracic Epidural Catheter Placement

Thoracic epidural analgesia provides reliable acute pain relief and reduces morbidity and mortality for patients undergoing abdominal or thoracic surgery. Conventional methods for thoracic epidural puncture relied on the surface anatomy of the spine, which often leads to incorrect identification of the targeted level and high failure rate, which limited the clinical application of epidural analgesia. The success rates of landmark-based lateral epidurals in the thoracic spine in our institution was around 75%. With the widespread use of ultrasound in regional anesthesia, widespread evidence supports neuraxial ultrasound assisting lumbar epidurals with great success, however the use of ultrasound for guidance in thoracic epidurals is still not widespread. David B. Auyong reported the first randomized study to evaluate the value of preprocedure ultrasound in thoracic epidural catheterization [14]. In this study, he found that preprocedural ultrasound did not significantly reduce the time required to identify the thoracic epidural space, but decreased the number of needle skin punctures and mean pain scores after surgery. Furthermore, Daniel J. Pak [12] described the successful real-time ultrasound-guided thoracic epidural technique in a paramedian sagittal oblique view. Recently, Karmakar and his colleagues reported a prospective, randomized superiority trial, which indicates that real-time ultrasound guidance is superior to a conventional anatomic landmark-based technique for first-pass success during TEP [15]. Since 2015, the author Fei Liu started to do ultrasound-guided epidural anesthesia replacing landmark based thoracic epidural anesthesia for postoperative pain in patients after abdominal surgery. A success rate beyond 90% was achieved [16].

### 5.2. Epidural Catheter Placement Was Achieved in Every Patient

In our center, preprocedure ultrasound scanning or real time ultrasound guidance has been a routine procedure for thoracic and lumbar epidural analgesia for 7 years. The author (the leader of the acute pain service) is skilled in real time ultrasound-guided epidural puncture. She found that some patients complained of severe pain postoperatively, which could not be relieved by strong opioids and NSAIDS. Thus, she thought that only epidural analgesia could help them, and after getting consent from the patients, she started to do real-time ultrasound guiding epidural punctures for them. During the procedure, strong pain prevents them from lying in a standard lateral position (neck to knee). Therefore, the author could only do epidural puncture in these patients lying in a comfortable lateral position. We found that although the patient’s position was not ideal for the traditional land mark epidural puncture, under ultrasound guidance we still could find the inter laminar space for needle advancement. Therefore, we started this preliminary study and tried to prove that ultrasound-guided epidural puncture could also be done for patients who could not use the standard knee to head position.

### 5.3. Under Ultrasound, the Vertebral Lamina, Intervertebral Space and Posterior Complex in the Paramedian Oblique Sagittal View Were All Good at the T6-L1 Level in the Patients without “Forehead to Knees”

In the 60 patients in this study, in the paramedian oblique sagittal view, the visibility of the vertebral lamina, intervertebral space and posterior complex under ultrasound were as high as 93.33%, 81.67% and 70.00%, respectively. The anterior complex and dural sac were poorly visible because of the extreme caudad angulation of the spinous processes and the overlapping laminae. However, we measured the width of the interlaminar space, which was 0.70 ± 0.23 cm at T6-L1, which was much narrower compared to the lumbar spine interlaminar space. Although it was very narrow, the good visibility of the intervertebral space provided an ideal route for epidural needle trajectory. This is consistent with the results of previous literature [8,17,18,19]. However, it is important to note that all of the articles presented were lumbar epidural punctures. Only ShengJin Ge et al. reported one case of an ultrasound-assisted paramedial-lateral approach of thoracic epidural puncture (T 7-8) [20]. However, this report could not provide information on the success rate of the puncture, catheterization, and adverse reactions. T Grau also showed that, despite the limited capacity of US to depict the thoracic epidural space, US proved to be of better value than MRI in the depiction of the dura mater [21]. These studies suggest that ultrasound-assisted thoracic epidural puncture is acceptable. This study not only provides a series of information about thoracic epidural puncture under ultrasound, but also puts forward the concept of comfort. It aims to provide patients with sufficient therapeutic effects and comfortable care at the same time.

### 5.4. Epidural Catheterization to Patients without Flexion of Knees and Neck under Ultrasound-Guided Has a High Success Rate and Accurate Analgesic Effect

Among the 60 patients observed, epidural puncture was performed in 59 patients with good visibility, except for one patient with had poor epidural visibility. A total of 55 patients completed the epidural catheterization successfully, with the exception of four patients who experienced puncture into the blood vessel or neuroparesthesia. The success rate of epidural catheterization was as high as 91.67%, and the postoperative analgesic effect was satisfactory for 98.18% of patients without nausea, pruritus and other discomfort. Thoracic epidural analgesia is a common method of pain relief for major thoracic and abdominal surgery [22]. However, due to the difficulty of the puncture in the past, the possible complications such as spinal cord injury and pneumothorax have limited its use [23]. T Grau et al. compared the ultrasound method with the traditional puncture method, and the results showed that the visualization procedure was very effective and reduced the injury caused by repeated puncture [24]. In addition, based on the visualization characteristics of ultrasound, the occurrence of the above complications can be avoided to a certain extent. Although previous studies showed that ultrasound guided epidural puncture in real time was feasible, it was still done when the patient was in an “ideal” position. Interestingly, Ban Tsui et al. performed on epidural space of male adult cadavers and found that ultrasound imaging and real-time ultrasound needle guidance for nerve blocks at the trunk and epidural space can be used in “stiff” cadavers [25]. This provides strong evidence for conducting epidural puncture in the comfortable position. Thus, our study suggests that visual manipulation makes epidural catheterization not only easier to perform, but also reduces the requirements of the procedure. This allows for epidural catheterization in a comfortable position for patients who are unable to provide a strict knee and neck flexion position.

### 5.5. Limitations

This study was only an observational one, and there was a lack of a control variable. In addition, there were certain limitations in the selection of patients. In the next step, we plan to include a wider range of people (such as patients with obesity, the elderly, and pregnant women), to start a prospective randomized controlled study to compare the success rate between the comfortable position and the strict knee and neck flexion position.

## 6. Conclusions

Under the comfortable position on the left lateral decubitus, the success rate of epidural catheterization under the guidance of real-time ultrasound was as high as 91.67%, and the analgesic efficiency was 98.18%. This technique is feasible for patients who are unable to achieve the forehead to knees position, while improving the comfort of patients during treatment.

## Figures and Tables

**Figure 1 jcm-11-06459-f001:**
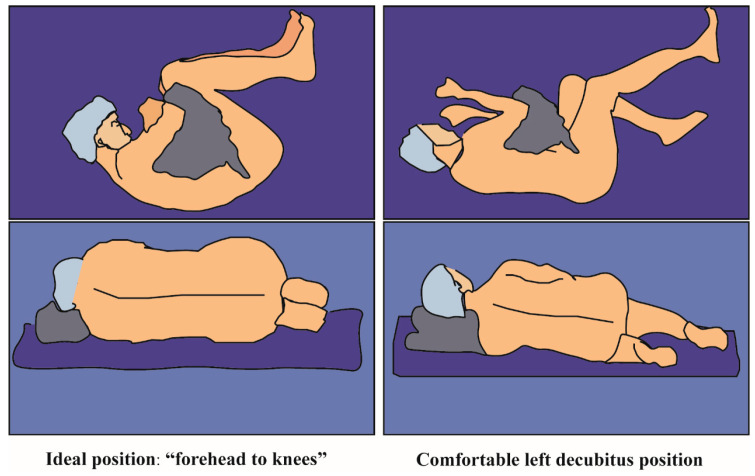
Patient’s position during the epidural catheter puncture. (**Left**): Ideal position (“forehead to knee”). (**Right**): Comfortable position (The patient wasn’t required to flex their thighs on their abdomen and flex their neck).

**Figure 2 jcm-11-06459-f002:**
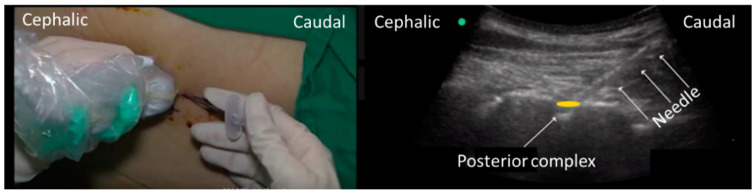
Example of how to hold the probe (**left the probe was firmly held in the marked position by the operator’s left hand****. A 18G Tuohy needle from the caudal of the probe was advanced to by the operator’s right hand**). Ultrasound imaging while operating puncture (**right**) THE needle was seen in the ultrasound image.

**Figure 3 jcm-11-06459-f003:**
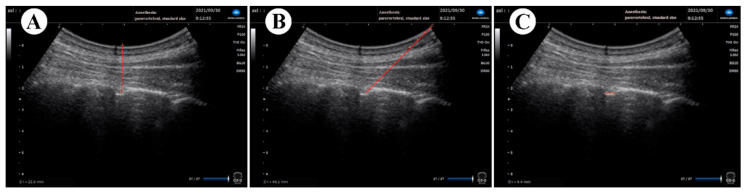
Example of ultrasonic image measurement in comfortable position. Vertical (**A**) and oblique distances (**B**) from the skin to the posterior dura and width of the interlaminar space were also measured (**C**) using the relevant “frozen image” of the ultrasound.

**Figure 4 jcm-11-06459-f004:**
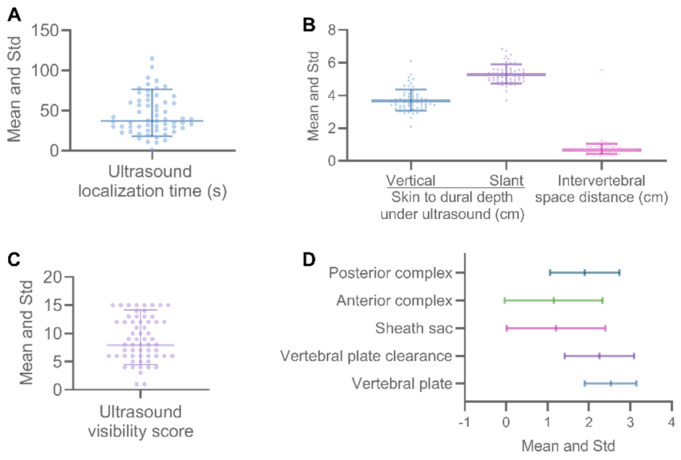
Quantitative measurements and ultrasound visibility score of neuraxial structures. The ultrasound localization time of all patients was under 2 min (120 s) (**A**: ultrasound localization time (s)). The vertical distance from skin to dura measured by ultrasound was about 3.75 cm (**B**, left), and the slope distance was about 5.62 cm (**B**, right). Different transverse nerve structures were shown in different states under ultrasound (**D**), and the total UVS score ranged from 1 to 15 points (**C**). Abbreviation: % = percentage, Std = Standard deviation, s = second(s); cm = centimeter(s).

**Figure 5 jcm-11-06459-f005:**
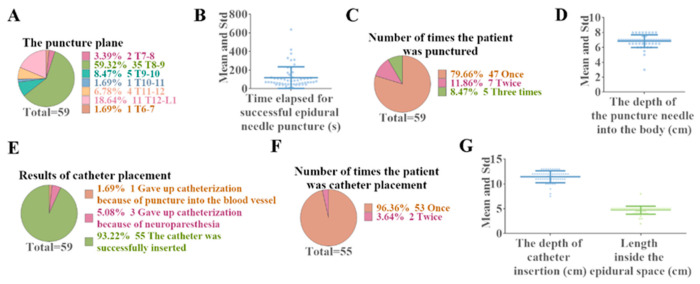
Outcomes of ultrasound-guided epidural catheterization in the absence of specific postures. Of the 60 cases observed, only one patient abandoned epidural puncture due to poor ultrasound imaging results. Among the 59 patients with successful puncture, the patients with puncture plane at T8-9 accounted for the largest proportion (**A**: Puncture plane distribution of successful epidural puncture), and the average time was 119.37 s (**B**: Duration of epidural puncture). The majority of patients completed the epidural catheterization procedure successfully the first time (**C**: Number of epidural punctures performed by the patient), and the average depth of puncture needle placement was 6.83 cm (**D**: The depth of the epidural puncture needle (cm)). Of the patients with successful puncture, 55 completed epidural catheterization (**E**: Results of catheter placement), of whom 96.36% completed the catheterization on the first attempt (**F**). The mean depth of catheter placement (**G**, Left) and the mean depth in the epidural space (**G**, Right) were 11.41 cm and 4.72 cm, respectively. Abbreviation: % = percentage, T = Thoracic vertebra, L = lumbar vertebra, Std = Standard deviation, s = second(s); cm = centimeter(s).

**Table 1 jcm-11-06459-t001:** Basic characteristics of patients.

Parameter		Numerical Information
Gender	Male, n, %	41, 68.33%
	Female, n, %	19, 31.67%
Age (years-old)	Mean ± Std	57.44 ± 11.36
	Min~Max	29–80
	Md (IQR)	59.00 (18.00)
	≥ 60 years, n, %	29, 49.15%
Height (cm)	Mean ± Std	163.91 ± 6.82
	Min~Max	150.00~180.00
	Md (IQR)	165.00 (11.00)
Weight (kg)	Mean ± Std	59.30 ± 9.26
	Min~Max	42.00~82.50
	Md (IQR)	58.25 (11.75)
BMI (kg/m^2^)	Mean ± Std	21.88 ± 3.07
	Min~Max	18.00–30.00
	Md (IQR)	21.63 (4.17)
	≥24 kg/m^2^	12, 21,82%
Hypertension, n, %		4, 6.67%
Abnormal ECG, n, %		2, 3.33%
Chronic bronchitis, n, %		4, 6.67%

Abbreviation: n = number(s), % = percentage, Std = Standard deviation, Min = Minimum value, Max = Maximum value, Md = Median, IQR = Interquartile range, cm = centimeter(s), kg = kilogram(s), m = meter(s), ECG = Electrocardiograph.

**Table 2 jcm-11-06459-t002:** Ultrasound Visibility Score of Neuraxial Structures.

	0 Point	1 Point	2 Point	3 Point	Mean	Std	Median	IQR
vertebral lamina	0 (0.00%)	4 (6.67%)	21 (35.00%)	35 (58.33%)	2.52	0.62	3.00	1.00
intervertebral space	2 (3.33%)	9 (15.00%)	21 (35.00%)	28 (46.67%)	2.25	0.84	2.00	1.00
dural sac	26 (43.33%)	7 (11.67%)	16 (26.67%)	11 (18.33%)	1.20	1.19	1.00	2.00
Anterior complex	26 (43.33%)	10 (16.67%)	13 (21.67%)	11 (18.33%)	1.15	1.18	1.00	2.00
Posterior complex	3 (5.00%)	15 (25.00%)	27 (45.00%)	15 (25.00%)	1.90	0.84	2.00	1.25

Abbreviation: % = percentage, Std = Standard deviation, IQR = Interquartile range. Note: 0 point = not visible, 1 point = hardly visible, 2 point = well visible, 3 point = very well visible.

## Data Availability

The data presented in this study are available on request from the corresponding author.

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
