# Peer review of "Real Time Ultrasound-Guided Thoracic Epidural Catheterization with Patients in the Lateral Decubitus Position without Flexion of Knees and Neck: A Preliminary Investigation"

_jcm, 2022, doi:10.3390/jcm11216459_

Round 1
Reviewer 1 Report
Dear Authors,
It’s my pleasure to review this well-written manuscript, on a actual toipic.
The topic is really actual and the manuscript, well-written, explained a valid option for checking the correct position of thoracic epidural catheterization.
However, some comments need to be added:
- All the catheterizations were performed by the same anesthetist?
- How do you calculate the sample size?
- Did you record any severe adverse effects?
Thanks a lot
Author Response
Dear Professor:
Thanks very much for your advice and question. We have revised them one point by one point.
- Q: All the catheterizations were performed by the same anesthetist?
R: Yes, all the catheterizations were performed by Fei Liu, the corresponding author.
- Q: How do you calculate the sample size?
R: According to our institution’s previous experience with ultrasound-guided catheterization, the success rate is approximately 90%. So, when we calculated the sample size, we assume the success rate is 90%, the confidence interval width is 16%, the allowable error range is 8%, the dropout rate is 10%, and α = 0.05. By using PASS 2021 (v21.0.3), a sample size of 54 were calculated, and we choose 60 patients as the final sample size.We have added this in the article.(L139-144)
- Q: Did you record any severe adverse effects?
R:Yes, we have recorded all the adverse effects related to epidural puncture. Mild adverse effects inluded dural puncture, bloody tap, electric shock feeling during needle insertion and severe adverse effects contained total spinal anesthesia, epidural hematoma or abscess related to epidural puncture. (L125-128, L250-252)
Reviewer 2 Report
Review
L32: “ to be more human”: please use a different phrase to convey the message. This one is not clear in english.
L89: please change “ left side” by “left lateral decubitus”
L90-100: Although it seems to be implicit that the right paramedian approach was attempted, please describe it and mention that the epidural needle was directed in a trajectory caudad to cephalad and lateral to medial or otherwise the authors prefer. My point is to make clear for the reader the procedure.
Also please mention some sonoanatomical
elements for the benefit of the reader.
L97: please mention clearly when the Ultrasound probe is left aside and the two hands of the single operator continue the procedure with LOR to air. Please include (LOR: loss of resistance)
L101-102: the numbering “1)” is missing.
L104-106: Regarding the Ultrasound structures, conservatively the dural sac is assumed by the delimitation of the anterior and posterior complex rather than visualized directly. And the spinal cord is not really visualized directly. Sometimes some pulse can be appreciated but it can not really attribute to spinal cord or epidural space vessels.
I suggest to define and describe: vertebral lamina, intervertebral space, posterior complex (ligament flavum, epidural space and posterior dura mater), dural sac and anterior complex (anterior dura mater, posterior longitudinal ligament and posterior aspect of the vertebral body)
L130: please change “ left side” by “left lateral decubitus”
L131: please provide SD in addition to average for both measurements
L132; Please change “ cancelling” by “cancelled”
L169: The concept of ultrasonic position has not been defined. Do you refer to ultrasound localization?
L175-186: As mention above, I suggest to limit the description the structures of the ultrasound imaging.
Table 2: same as above.
L201-202: I suggest to change “twice” and “three times” by “ were punctured by the second attempt” and “punctures by the third attempt”.
L204-205: in Line 135 it is mentioned 3 transient paresthesia, which seems to be in contrast to this 1 neuroparesthesia. Do you refer paresthesia with the catheter rather than 3 paresthesia with the needle. Please clarify and make it clear in the text.
Figure 4: Graph E “Results of Catheter plcement” differs from the information in the text regarding neuroparesthesia.
L213: please change “operation” to epidural procedure” to distinct from surgical operation/procedure.
L223-224: Please clarify the information of neuroparesthesia 1 or 3. It seems that the catheter placement was completed despite neuroparesthesia in 3 cases and it was cancelled in 1 case.
L255: Please change “neuropathy” to “paresthesia”
L265: please change “lever” to “level”
L267: please change “lots of evidences supported” to “widespread evidence supports”
L274-275: please change to “ describe the successful real-time ultrasound-guided thoracic epidural technique in a paramedian sagittal oblique view.”
L279: please change “guiding” to “guided”
L284: please change to “During the procedure,”
L287: please change to “ could find the inter laminar”
L289: please change to “ tried”
L290: please change to “ guided”
L343: change from “left side” to left lateral decubitus”
Final comment: readers would benefit from a more detailed description of the real time US-guided procedure.
Author Response
Dear Professor:
Thanks very much for your advice and question. We have revised them one point by one point.
1.Q: L32: “ to be more human”: please use a different phrase to convey the message. This one is not clear in english.
R: L34: “ to be more human” → “to improve patients’ comfort”.
2.Q: L89: please change “ left side” by “left lateral decubitus”.
R: L93: “ left side” → “left lateral decubitus”.
3.Q: L90-100: Although it seems to be implicit that the right paramedian approach was attempted, please describe it and mention that the epidural needle was directed in a trajectory caudad to cephalad and lateral to medial or otherwise the authors prefer. My point is to make clear for the reader the procedure. Also please mention some sonoanatomical elements for the benefit of the reader.
R: We have already attached a picture of real-time ultrasound guidance (Figure 2), and uploaded a video that the corresponding author Fei Liu introduced the ultrasound image of spine and the complete process of real-time ultrasound-guided epidural puncture.
4.Q: L97: please mention clearly when the Ultrasound probe is left aside and the two hands of the single operator continue the procedure with LOR to air. Please include (LOR: loss of resistance)
R: L94-L108, and (Figure 2 Left).
5.Q: L101-102: the numbering “1)” is missing.
R: L113: the numbering “1)” has already been added.
6.Q: L104-106: Regarding the Ultrasound structures, conservatively the dural sac is assumed by the delimitation of the anterior and posterior complex rather than visualized directly. And the spinal cord is not really visualized directly. Sometimes some pulse can be appreciated but it can not really attribute to spinal cord or epidural space vessels. I suggest to define and describe: vertebral lamina, intervertebral space, posterior complex (ligament flavum, epidural space and posterior dura mater), dural sac and anterior complex (anterior dura mater, posterior longitudinal ligament and posterior aspect of the vertebral body).
R: The name of structures have been changed to much better form (vertebral lamina, intervertebral space, dural sac, anterior complex and posterior complex).
(L22-23, L116-117, L195-200, Table 2, L258, L260, L270, L309, L313, L315, L319)
7.Q: L130: please change “ left side” by “left lateral decubitus”
R: L147: “ left side” → “left lateral decubitus”
8.Q: L131: please provide SD in addition to average for both measurements
R: L148-149: “Time of pre-procedure scanning and puncture was 44.78 ± 24.81 s and 119.37 ± 116.70 s”.
9.Q: L132: Please change “ cancelling” by “cancelled”
R: L150: “ cancelling” → “cancelled”.
10.Q: L169: The concept of ultrasonic position has not been defined. Do you refer to ultrasound localization?
R: Yes, we referred to ultrasound localization, and we have corrected the expression in our article.(Figure 4 A, L187-188)
11.L175-186: As mention above, I suggest to limit the description the structures of the ultrasound imaging.
R: L195-200: The name of structures have been changed to much better form (vertebral lamina, intervertebral space, dural sac, anterior complex and posterior complex).
12.Q: Table 2: same as above.
R: Table2: The name of structures have been changed to much better form (vertebral lamina, intervertebral space, dural sac, anterior complex and posterior complex).
13.Q: L201-202: I suggest to change “twice” and “three times” by “ were punctured by the second attempt” and “punctures by the third attempt”.
R: L218-219: “ twice” and “three times” → “ were punctured by the second attempt” and “punctures by the third attempt”
14.Q: L204-205: in Line 135 it is mentioned 3 transient paresthesia, which seems to be in contrast to this 1 neuroparesthesia. Do you refer paresthesia with the catheter rather than 3 paresthesia with the needle. Please clarify and make it clear in the text.
R: It could be a clerical error in the manuscript, the procedure stage we referred to is catheter placement. In order not to cause ambiguity, we delete the text here (previous manuscript L204-205), and we describe the data at L238-240.
15.Figure 4: Graph E “Results of Catheter placement” differs from the information in the text regarding neuroparesthesia.
R: Same as above.
16.Q: L213: please change “operation” to epidural procedure” to distinct from surgical operation/procedure.
R: L228-229: “operation” → “epidural procedure”
17.Q: L223-224: Please clarify the information of neuroparesthesia 1 or 3. It seems that the catheter placement was completed despite neuroparesthesia in 3 cases, and it was cancelled in 1 case.
R: L238-240: Sixty patients were included in our study, one of 60 patients gave up puncture due to poor ultrasound imaging results. Fifty-nine patients got successful puncture procedure, one of 59 patients failed to complete the catheter placement for puncture into the blood vessel, and three of 59 patients failed to complete the catheter placement for neuroparesthesia. So, there were 55 patients who successfully completed the catheter placement.
18.Q: L255: Please change “neuropathy” to “paresthesia”
R: L270: “neuropathy” → “paresthesia”
19.Q: L265: please change “lever” to “level”
R: L280: “lever” → “level”
20.Q: L267: please change “lots of evidences supported” to “widespread evidence supports”
R: L283: “lots of evidences supported” → “widespread evidence supports”
21.Q: L274-275: please change to “ describe the successful real-time ultrasound-guided thoracic epidural technique in a paramedian sagittal oblique view.”
R: 290-291: it has been changed to “describe the successful real-time ultrasound-guided thoracic epidural technique in a paramedian sagittal oblique view.”
22.Q: L279: please change “guiding” to “guided”
R: L301: “guiding” → “guided”
23.Q: L284: please change to “During the procedure,”
R: L305: it has been changed to “During the procedure,”
24.Q: L287: please change to “ could find the inter laminar”
R: L309-310: it has been changed to “could find the inter laminar”
25.Q: L289: please change to “ tried”
R: L311: it has been changed to “ tried”
26.Q: L290: please change to “ guided”
R: L312: it has been changed to “guided”
27.Q: L343: change from “left side” to “left lateral decubitus”
R: L367: it has been changed to “left side” to “left lateral decubitus”
Reviewer 3 Report
From your description of the inspiration for this study, you imply that lateral thoracic epidurals performed in the "comfortable" lateral position have been performed with some frequency at your institution. Are you able to include success rates without real-time epidural guidance prior to its widespread use at your institution?
Would you be able to include success rates for landmark based sitting or lateral epidurals for your institution/patient population?
The 5 patients that did not have successful placement are described as being a vascular puncture (presumably an epidural vein) or transient paresthesias. At many institutions, clinicians would consider these events a reason to redirect the needle or place the epidural at a different level. Is it common practice at your institution to completely abort an epidural if a paresthesia occurs or if an epidural catheter ends up in a vein?
Author Response
Dear Professor:
Thanks very much for your advice and question. We have revised them one point by one point.
- From your description of the inspiration for this study, you imply that lateral thoracic epidurals performed in the "comfortable" lateral position have been performed with some frequency at your institution. Are you able to include success rates without real-time epidural guidance prior to its widespread use at your institution?
R: This question is answered in Q2.
- Would you be able to include success rates for landmark based sitting or lateral epidurals for your institution/patient population?
R:The success rate of epidural puncture is affected by several kinds of factors: the qualification of anesthetist, the puncture level (the lumbar spine is much easier than thoracic spine), age, weight, height, whether the patients have rachiterata, etc. We have not yet caculated the precise data, the success rate could be approximately 75% based on the interview of a few senior anesthesiologists(L281-282). Subsequently, we will perform RCTs to compare the thoracic epidural puncture success rate between landmark method and ultrasound guidance.
- The 5 patients that did not have successful placement are described as being a vascular puncture (presumably an epidural vein) or transient paresthesias. At many institutions, clinicians would consider these events a reason to redirect the needle or place the epidural at a different level. Is it common practice at your institution to completely abort an epidural if a paresthesia occurs or if an epidural catheter ends up in a vein?
R: In routine work, if it happened to a vascular puncture or transient paresthesias, we would consider adjusting the direction of the needle or choose a different level. But as to protect the patients in this study, we didn’t try further, and stopped the puncture.